

# Robotic evaluation of a 3D-printed scaffold for reconstruction of scapholunate interosseous ligament rupture: a biomechanical cadaveric study

Alastair R.J. Quinn[1,2,3], Jayishni N. Maharaj[1,2,3], Randy Bindra[1,2,4,5], Amelia Carr[6], Natividad Gomez[6], Kaecee Fitzgerald[1,2], Nataliya Perevoshchikova[1,2,3], Cedryck Vaquette[6], Claudio Pizzolato[1,2,3], Minghao Zheng[7], David Lloyd[1,2,3] and David J. Saxby[1,2,3]

[1] Griffith Centre of Biomedical and Rehabilitation Engineering, Menzies Health Institute Queensland, Griffith University, Gold Coast, Queensland, Australia
[2] Advanced Design and Prototyping Technologies Institute, Griffith University, Gold Coast, Queensland, Australia
[3] School of Health Sciences and Social Work, Griffith University, Gold Coast, Queensland, Australia
[4] School of Medicine and Dentistry, Griffith University, Gold Coast, Queensland, Australia
[5] Department of Orthopaedic Surgery, Gold Coast University Hospital, Gold Coast, Queensland, Australia
[6] School of Dentistry, University of Queensland, Brisbane, Queensland, Australia
[7] Medical School, Faculty of Health and Medical Sciences, University of Western Australia, Nedlands, Western Australia, Australia

Corresponding author
Alastair R.J. Quinn,
alastair.quinn@griffith.edu.au

## ABSTRACT

**Background**. Rupture of the scapholunate (SL) interosseous ligament (SLIL) is a challenging injury to treat surgically due to the small and complex nature of the SL linkage. This study was a preliminary robotic assessment of the immediate biomechanical effects of a novel 3D-printed scaffold used to reconstruct the ruptured SLIL.

**Methods**. Nine minimally loaded cadaveric wrists underwent robotically manipulated flexion-extension and radial-ulnar deviation under conditions of intact, transected, and reconstructed SLIL. Simulated radiographic measures (*i.e.*, SL angle and SL gap) and three-dimensional SL gap across wrist motions were used to assess static and dynamic stability of the reconstructed SLIL.

**Results**. Three cadaveric specimens produced complete results across all experimental conditions. Intact SL linkage had a SL angle comparable (but slightly lower) than normative literature values. Once the native SLIL was transected, SL angle disruption was evident, and largely restored once the scaffold was surgically installed. Similar results were seen for SL gap. Results of the dynamic three-dimensional SL gap indicated the scaffold restored dynamic stability to a limited extent.

**Conclusion**. Static and dynamic stability of the SL linkage was not compromised by surgical installation of the scaffold. Scaffold installation provided limited restoration of SL linkage towards native values; however, the small number of cadaveric specimens and minimal articular loading applied to the radiocarpal joint limits generalization. Overall, the scaffold may provide adequate mechanical fixation of the SL linkage and enable biological ingrowth of ligament.

## INTRODUCTION

Hyperextension of the wrist combined with moderate to large compressive loading can cause rupture of the scapholunate (SL) interosseous ligament (SLIL) (*Mayfield, Johnson & Kilcoyne, 1980*), which is a common wrist injury sustained by both young athletes and the elderly. Following SLIL rupture wrist function deteriorates precipitously (*Laulan, Marteau & Bacle, 2015*). Instability at the SL joint can be graded by the gap between the bones assessed through arthroscopy (*Geissler, 2013*) or radiographic appearance of progressive arthritis (scapholunate advanced collapse pattern) (*Watson & Ballet, 1984*). Reconstruction of the SL linkage is reserved for early stages of the disease before degeneration of the articular surfaces.

Various types of surgical reconstruction techniques have been described based on duration since injury, severity of carpal collapse on imaging and presence of degeneration (*Kitay & Wolfe, 2012*; *Andersson, 2017*; *Wahed et al., 2020*; *Bakker et al., 2022*). Current surgical techniques for management of acute SLIL rupture include sutures with or without anchors, reduction and temporary pinning of the carpal bones or tendon autograft reconstruction. Despite many options, there is no clinical consensus on best treatment (*Herbert, 1991*; *Linscheid & Dobyns, 1992*; *Holmes et al., 2017*; *Wessel & Wolfe, 2023*), and existing techniques are limited by high rates of re-rupture, poor mechanical performance, and wrist stiffness (*Rohman et al., 2014*). These challenges highlight the urgent need for a novel and effective technique for reconstructing the failed native SL linkage.

Wrists with a ruptured SLIL show marked differences in their SL kinematics compared to wrist with intact SLIL (*Short et al., 2007*; *Short et al., 1995*). However, carpal kinematics vary considerably across individuals performing dynamic and static tasks, thus making establishment of normative, and hence pathological, kinematics challenging (*Short et al., 1995*; *Ruby et al., 1987*; *Falck Larsen, Mathiesen & Lindequist, 1991*; *Rainbow et al., 2008*; *Schernberg, 1990*; *Watson & Black, 1987*). Varying study methods (*i.e.,* data acquisition techniques, wrist pose/motion, data processing) also hinder establishment of normative kinematics (*Larsen et al., 1991*; *Shaw et al., 2019*; *Said et al., 2018*; *Lee et al., 2011*; *Dornberger et al., 2015*; *Pliefke et al., 2008*). Given the variability in carpal kinematics and challenges associated with establishing normative kinematics, evaluation of clinical solutions used to treat the ruptured SL ligament is often limited to binary outcomes (*e.g.,* technique failed or not) or subjective rating of pain and dysfunction.

Recently, a novel SLIL scaffold has been proposed (*Gómez-Cerezo et al., 2021*; *Perevoshchikova et al., 2021*; *Lui et al., 2019*; *Lui et al., 2021*; *Carr et al., 2018*; *Lui, Vaquette & Bindra, 2017*), which is manufactured through 3D printing of medical-grade polycaprolactone (PCL) (*Lui et al., 2019*; *Lui, Vaquette & Bindra, 2017*). This scaffold is intended to provide immediate (*i.e., time zero to early stages of rehabilitation*) mechanical support to the SL joint. In animals, the scaffold has been shown to assimilate biologically,

with native tissue invading the scaffold bone plugs (*i.e.,* bone ingrowth) and bridging mid-section (*i.e.,* non-mineralized connective tissue) (*Lui et al., 2021*; *Lui et al., 2019*). What remains is to determine if this novel scaffold can restore human SL kinematics as assessed using standard clinical measures as well as more advanced dynamic measurements.

The kinematics of the carpal bones vary substantially across individuals and are challenging to model *in silico*. However, applying appropriate physiological boundary conditions to a SLIL implant for testing is feasible through *in vitro* methods (*e.g.,* robotic control). The purpose of this study was to characterize SL linkage in cadaveric wrists with intact, transected, and reconstructed SLIL respectively. The reconstruction was performed using the novel scaffold, precision installed with bespoke surgical guides. Carpal kinematics were assessed under static and dynamic wrist conditions analogous to early-stage rehabilitation (*i.e.,* ~10 N radiocarpal compressive loading). Notably, SLIL rupture is identified clinically by measuring SL angle and gap with static radiographs (*Falck Larsen, Mathiesen & Lindequist, 1991*; *Chen et al., 2022*). However static radiographic measures alone cannot identify cases of dynamic instability (*Taleisnik, 1988*; *Boutin et al., 2014*), therefore, cadaveric-robot experiments were conducted to measure time-varying carpal bone kinematics and compute *in silico* clinical measures using digital models of the bones. Baseline characteristics of the intact SL linkage were defined for each wrist specimen, followed by acute (*i.e.,* transected SLIL), and finally reconstructed (*i.e.,* installed scaffold) conditions.

## MATERIALS & METHODS

In this study, we used robotic manipulation of cadaveric wrists to study carpal kinematics when the SLIL was intact, transected, and reconstructed using a novel scaffold. This study was approved by the institution's Human Research Ethics Committee (GU Ref No: 2018/533). Written informed consent for body donation and use in research was obtained from all donors prior to death, in accordance with institutional and ethical guidelines. Nine fresh frozen cadaveric specimens harvested proximal to the elbow with normal wrist radiographs (all males, 59–75 years of age, 46–70 kilograms, 1.62–1.76 m of stature) were included in this study. Three specimens (65–74 years of age, 46–62 kilograms, 1.72–1.76 m of stature) successfully completed the full experimental protocol (described in detail below). Prior to each cadaveric experiment session, an anatomical model of the specimen was created to plan the surgery (*i.e.,* physically practice scaffold installation) and refine experimental instruments.

### Pre-experiment testing with anatomical models

Each specimen required a personalised scaffold, adjusted for size and installation location. This was designed digitally and validated through physical testing of the associated anatomical model (Fig. 1). The advantage of digital design followed by physical testing using anatomical models prior to cadavers is it is relatively inexpensive and fast to engage in design iterations. For each cadaveric specimen, 3D surface mesh models of scaphoid, lunate, capitate, 3rd metacarpal, and distal radius were generated from segmented computerized tomography (CT) or magnetic resonance imaging (MRI) scans using Mimics Innovation

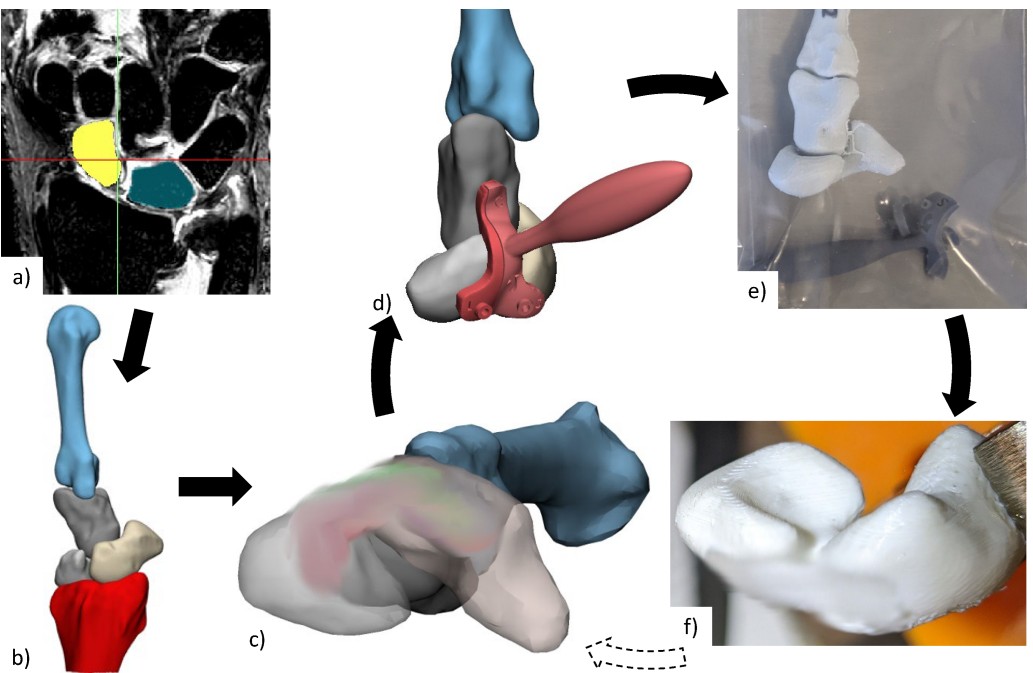

**Figure 1   Anatomical testing workflow.** (A) CT or MRI scans of each specimen are segmented to create (B) 3D reconstructed bones, from which (C) personalised scaffold (size and placement) was chosen and (D) surgical guides designed. (E) Anatomical bones, scaffold, and surgical installation guides were 3D printed for (F) practice of surgical installation. The scaffold has been blurred in all images, due to a pending patent application. CT, computed tomography; MRI, magnetic resonance imaging; 3D, three-dimensional.

Suite (Materialise, Leuven, Belgium). MRI scans were used when CT was of poor quality. No specimens showed any signs of SLIL damage (*i.e.,* SL gap > three mm or surrounding tissue abnormalities) from imaging.

Some cadaveric specimens were not completely thawed to room temperature, resulting in medical images and their associated 3D mesh models containing slightly flexed metacarpals. An articulating statistical shape model was then used to register (*i.e.,* rigidly transform) each set of bones into a neutral wrist position (*Zhang et al., 2014*; *Grant et al., 2020*). Scaffold installation location and size was determined by an orthopaedic surgeon specializing in hand and wrist surgery (co-author R.B.). The surgical installation guides were then generated based on scaffold size and 3D meshes for each specimen.

The scaffold and anatomical models for each cadaveric specimen were 3D-printed in PCL and Polylactic Acid (PLA) respectively, using Filament Deposition Modelling technology (3D-Bioplotter Developer Series from EnvisionTEC, Germany for the scaffold and Flashforge Inventor 3D Printer, Flashforge 3D Technology, Jinhua City, China for the models). Surgical installation guides (designed with Materialise 3-matic Medical, Leuven, Belgium) were 3D-printed using Stereolithography (Anycubic, Kowloon, Hong Kong) using 405 nm ultraviolet sensitive resin. These 3D-printed surgical guides were designed

with 1.1 mm guide wire holes to facilitate accurate placement (location and orientation) of holes for scaffold installation.

The 3D-printed anatomical models and surgical guides were then used by the surgeon for a simulated surgical installation to confirm size and placement of scaffold. If needed, installations were repeated with new prints generated from adjusted scaffold installation locations. Selected scaffold and surgical guides were manufactured again and used for the individual cadaver experiments.

## Experiment setup

For each experiment session, a specimen was thawed 24 h in advance, and testing lasted approximately 3-hours. Each cadaveric specimen was dissected (by co-author R.B.) to expose the dorsal surface of the scaphoid, lunate, capitate, and Lister's tubercle of the radius (Fig. 2). Active infra-red motion capture markers (~seven mm in diameter and ~6.5 mm tall cylinders, each containing six LEDs equally spaced in a circle, VZ10K, Phoenix Technologies Inc., Vancouver, Canada) were rigidly attached to scaphoid, lunate, 3rd metacarpal, and radius using threaded Kirschner wires and custom-made marker holders (Fig. S1). The wires were embedded into the cadaveric specimen using a specimen-specific placement guide to allow for surgical scaffold installation as well as avoid specimen-marker contact with wrist motion and avoid collision with neighbouring marker pins. The only secondary stabilizer of the SLIL that was transected was the dorsal radiocarpal ligament. Without transection, this ligament would collide with the marker pins and the surgical site would not be exposed for later surgical scaffold installation.

### Cadaver-robot interface

Each specimen was mounted into the robotic manipulator (Quinn et al., 2023) with the zero-reference set when the wrist was in neutral position (Fig. 3). Rigid attachment of the specimen to robot required custom made mounting fixtures (Fig. S2). Metacarpals two to four (for smaller specimens) or two to five (for larger specimens) were transfixed, using Steinmann pins, from one end of the mounting fixture through the metacarpals and through another hole on the other side. Then, one to two additional K-wires were passed through as many metacarpals the specimen allowed at varying angles and positions until fixation was ridged (Fig. S3). Forearm pronation and supination were restricted by fixing at least one Steinmann pin through radius, ulna, and lower mounting fixture holes while the coronal plane of the hand was parallel with the humerus. If needed, the forearm was transfixed with additional pins at varying angles and positions along the radius and ulna.

### Robotic manipulation of specimen

Each specimen was manipulated using a six-degrees of freedom gantry-gimbal robot. In this robot, translational mobilities are provided by near-frictionless (ride on air bearings) axes, while the $z$-axis (vertical) is near weightless (gravity counteracted by pneumatic pistons). Once the specimen was mounted in the robot, marker occlusion was assessed throughout all ranges of wrist motion. If markers were occluded at any point in the test trial, a mirrored design of the marker holder was used to reposition the markers and/or the tracking camera was repositioned. Then, the specimen was robotically manipulated

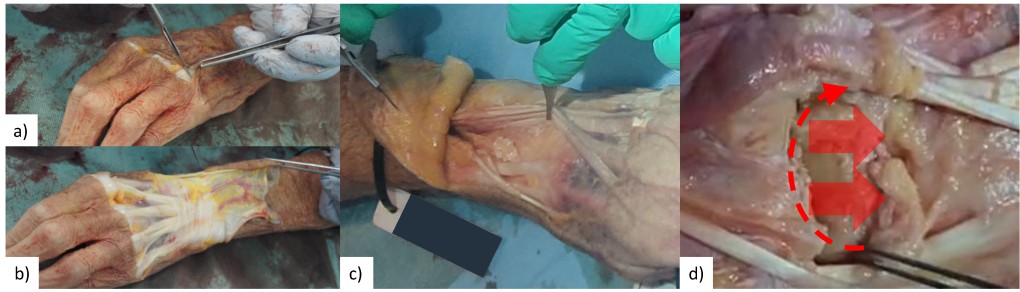

**Figure 2** **Surgical exposure of bone surface.** (A & B) Skin is removed from first knuckles (carpo-phalangeal joint) to radius neck. (C) Tendons of the extensor digitorum and extensor indicis were secured radially by sectioning extensor retinaculum in-between extensor pollicis longus and extensor indicis, and then suture remaining extensor retinaculum to tissue on radial side. (D) Section dorsal radiocarpal ligament from extensor pollicis longus along radiocarpal joint, then remove from scaphoid and lunate until roughly one cm of capitate is visible.

through flexion-extension (FE) for six cycles, followed by radial-ulnar deviation (RUD) for six cycles (Fig. 4). As the SLIL was intact at this stage (initiation of experiment), these tests are henceforth referred to as the "intact" configuration.

The next phase of testing involved transecting all parts of the SLIL (palmar, proximal, and dorsal) without unmounting from the robot, followed by repeating the same FE and RUD motions. These tests are henceforth referred to as the "transected" configuration. The mounting fixtures at each end of the specimen were then detached from the robot (maintaining specimen-fixture pinning) and the SLIL scaffold was surgically installed into the scaphoid and lunate bones after restoring their normal alignment. The specimen was reattached to the robot and manipulated through FE and RUD cycles. The latter tests are henceforth referred to as the "scaffold" configuration. Specimens were placed in a neutral position before beginning robotic manipulations, where the robot zero-reference was set.

## Determination of carpal kinematics

Motion capture combined with specimen-specific 3D mesh models were processed (Python 3.8) to model individual bone motions (Fig. 5). Motion capture data were first filtered to reduce noise and remove erroneous data points, then interpolated, and finally lowpass filtered (2nd order Butterworth filter with 0.5 Hz cut off frequency). Marker data registration (for each configuration) from global coordinates to model body coordinates was performed by aligning markers in neutral position to their designed positions based on marker pin placement guides. Motions of the scaphoid, lunate, and 3rd metacarpal, all relative to the radius, were computed using inverse kinematics (OpenSim 4.2 *Delp et al., 2007*; *Seth et al., 2018*).

Wrist motions were defined by the 3rd metacarpal relative to the radius (not by the robot gimbal), following standard conventions reported in literature (*Short et al., 2007*). A dynamic 3D measure of the SL gap across the motion cycles was generated using point kinematics, henceforth referred to as "3D SL gap". The two points used were generated from intersections of the respective SL bones with the scaffold digitally embedded atop

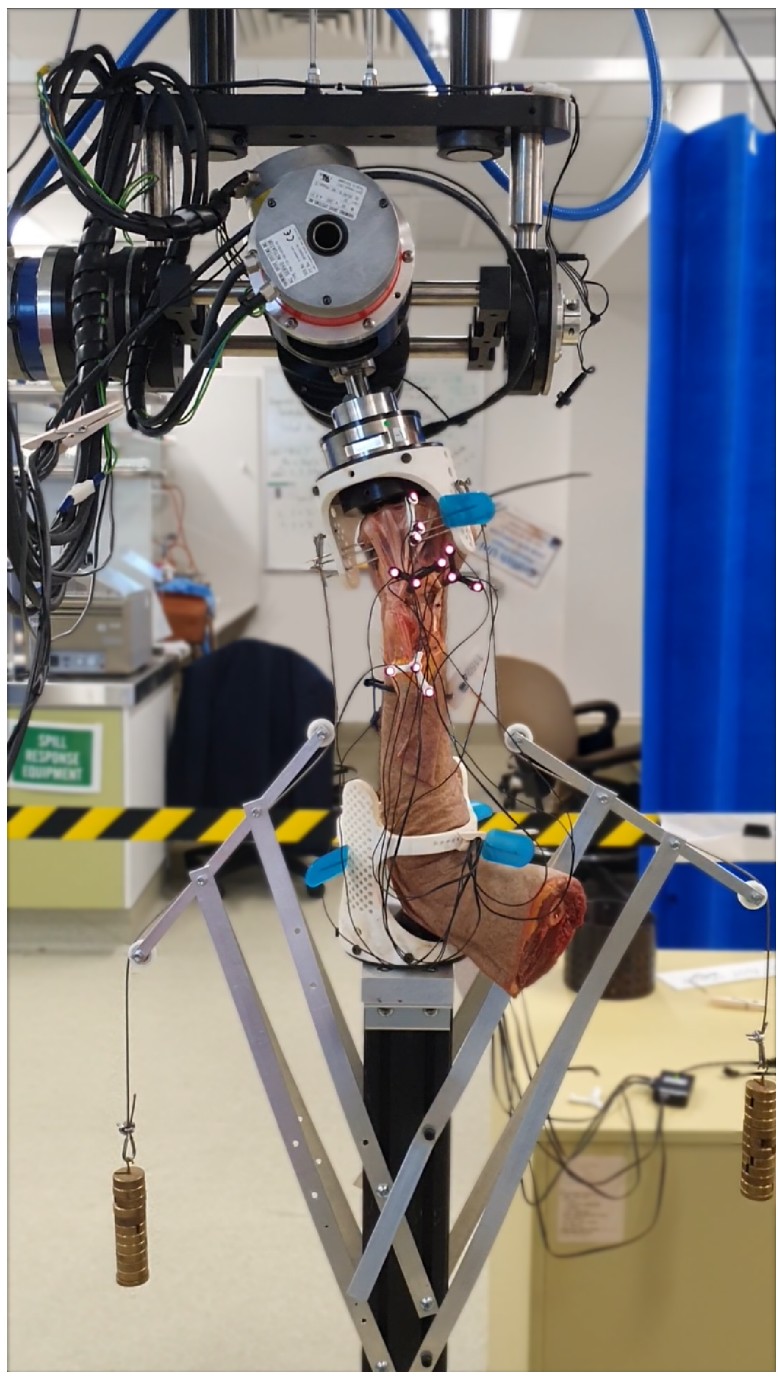

**Figure 3** **Cadaveric arm mounted in robot with weights (0.55 kg each) compressing radiocarpal gap.** Mounting fixture design accommodated different sized hand-forearm specimens, as the array of holes allowed fixation, achieved using Steinmann pins, from multiple directions.

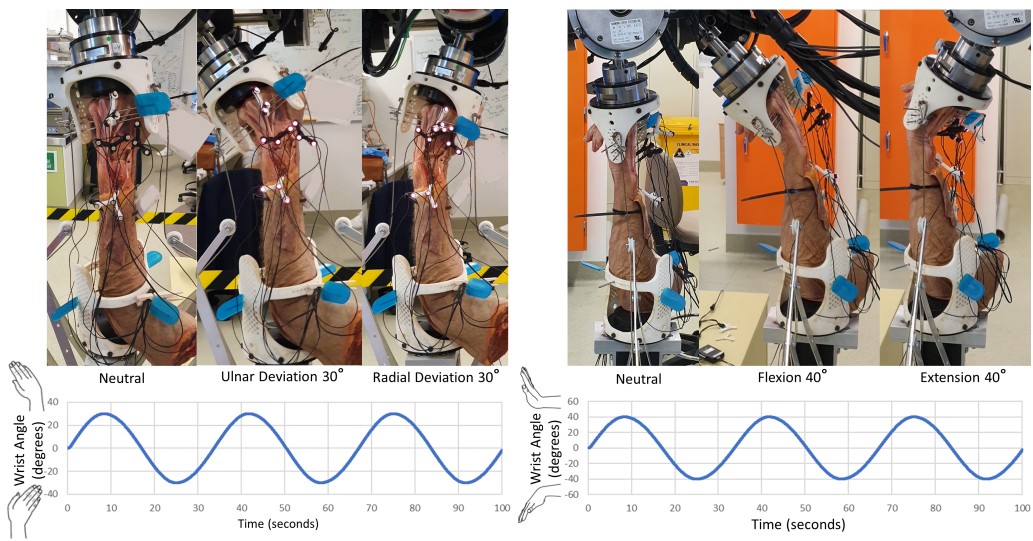

**Figure 4** **Manipulation profiles.** Velocity was limited to 10 °/s and acceleration to 10 °/s/s, this was to minimize any undesired loads resulting from the inertia of the robotic system. Motions were applied twice, consecutively, for data redundancy and in-case any components loosened. Left: radial-ulnar deviation ±30 °. Right: flexion-extension ±40 °.

the native SLIL footprint. These points were located on either side of the proximal SLIL (analogous to distal SL gap measure *Said et al., 2018*), and vary between specimens due to scaffold placement and anatomy. The 3D SL gap was computed for each specimen and each configuration across the wrist motion cycles. From these time-varying curves, discrete parameters were also extracted.

## Computation of radiographic measures

Kinematics combined with specimen 3D mesh data were processed (Python 3.8) to produce measures of SL alignment similar to those measured from standard radiographs. Posterior/anterior and lateral views of the bones were calculated using bony landmarks on the radius (Fig. S4) (*Badida et al., 2020*). For each specimen, the SL angle (*Falck Larsen, Mathiesen & Lindequist, 1991*) and SL gap (*Said et al., 2018*) were static metrics computed as the average of the measurements from the first frame (instant before specimens were dynamized) from all FE and RUD motion trials (four in total). These measurements correspond to static poses after 3, 6, and 9 motion cycles.

## RESULTS

### Surgical observations

The scaffold installation process was successful across all specimens, demonstrating efficacy of the procedure and validated the design geometry of the customised scaffolds and surgical installation guides. Upon completion of the robotic testing, all scaffolds remained securely installed and appeared fully intact once extracted. It was physically challenging to extract the scaffolds from both scaphoid and lunate, with the lunate being the easier. All cadaveric

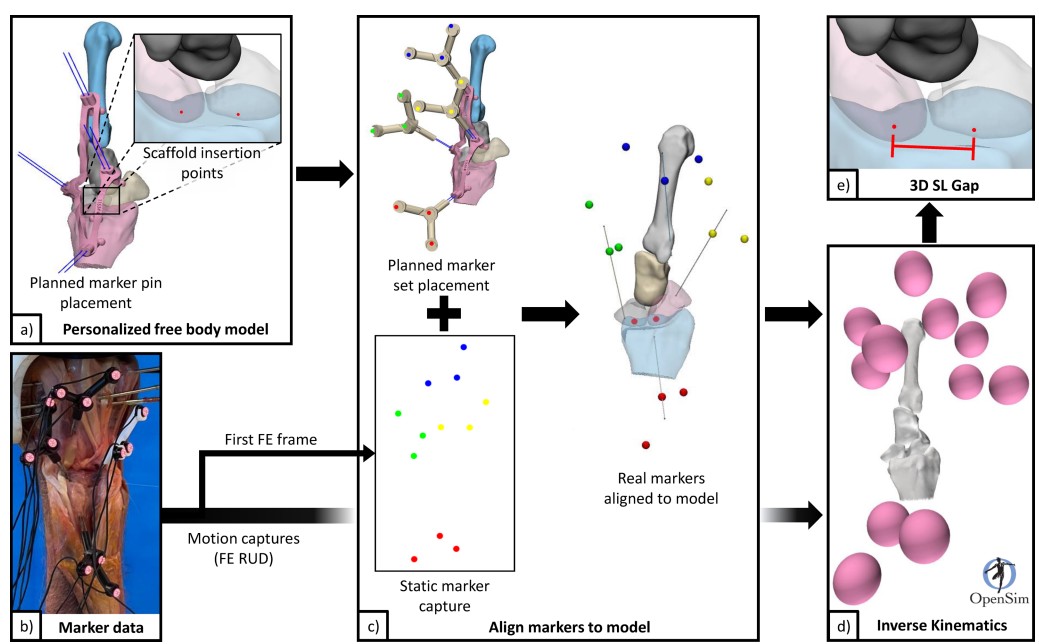

**Figure 5** **Data flow for processing pipeline.** (A) Personalised free body model from 3D surface meshes, and planned scaffold insertion points and marker pins. (B) Marker data from filtering raw motion capture. (C) Generate a personalized free body model with aligned markers using static motion capture data (first frame) and specimen specific free body model. (D) Generate kinematics for scaphoid, lunate, radius, and 3rd metacarpal from FE and RUD motion capture data. (E) Calculate 3D SL gap using point kinematics. 3D, three-dimensions; FE, flexion and extension; RUD, radial and ulnar deviation; SL, scapholunate.

specimens presented with fully intact SLIL in pre-experiment CT scans. Notably, during the dissection of specimen 5, it was discovered the SLIL was already ruptured along the proximal section, although the dorsal portion remained intact.

## Quantitative measures

Clinically relevant variables are shown in Fig. 6 along with derived statistics in Table 1. Of the experiments conducted, specimens 4, 6, and 7 completed all testing configurations. The 3D SL gaps from these valid experiments are shown in Fig. 7 along with their derived discrete variables in Tables 2 and 3. The other specimens encountered failures in bone quality holding marker pins and issues surrounding technical occlusion of markers due to the confined space. Indeed, marker pins were relocated for specimens 2, 3, 5, 8, and 9 to facilitate installation of scaffold, resulting in invalid motion capture data in these instances. Markers for specimen 3 fell out during transection of the SLIL. Specimen 1 had irreparable motion capture artifacts only present during motion, and specimen 7 scaffold configuration had no measurable SL gap (*i.e.*, bone edges abutting each other).

Supplementary data includes the kinematics of scaphoid, lunate, and scaphoid relative to lunate for specimens 4, 6, and 7, under all testing configurations (Fig. S5). Also included in the supplementary data are tabulated minimum values, maximum values, delta (maximum minus minimum), and values at neutral pose for wrist (3rd metacarpal relative to radius)

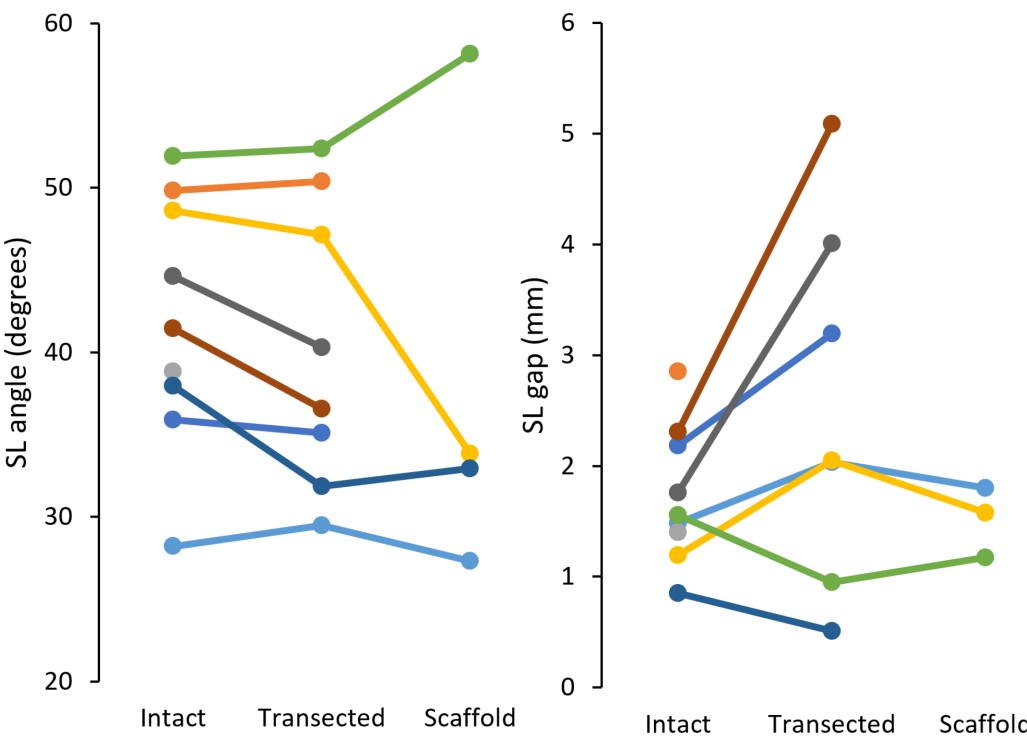

**Figure 6  Radiographic measures of SL angle (left) and SL gap (right) for each specimen (9 total).** SL, scapholunate.

**Table 1  Mean, range, standard deviation, and M ± 2 SD of radiographic measures.**

| Measure | Configuration | n | Mean | Range | SD | M − 2SD | M + 2SD |
|---|---|---|---|---|---|---|---|
| SL angle (°) | Intact | 9 | 41.9 | 28.2–52.0 | 7.2 | 27.6 | 56.3 |
| | Transected | 8 | 40.4 | 29.5–52.4 | 8.1 | 24.2 | 56.6 |
| | Scaffold | 4 | 38.1 | 27.3–58.1 | 11.9 | 14.4 | 61.8 |
| SL gap (mm) | Intact | 9 | 1.73 | 0.85–2.85 | 0.58 | 0.57 | 2.90 |
| | Transected | 7 | 2.55 | 0.51–5.09 | 1.52 | −0.50 | 5.59 |
| | Scaffold | 3 | 1.52 | 1.17–1.80 | 0.26 | 1.00 | 2.04 |

**Notes.**

SL, scapholunate; SD, standard deviation; M, mean.

motions and motions (x, y, z-axis rotations) of scaphoid, lunate, and scaphoid relative to lunate (Tables S1 to S8).

## DISCUSSION

In this study, a novel scaffold for reconstruction of the ruptured SLIL was assessed under static and dynamic conditions of robotically manipulated cadaveric wrists using current standard clinical measures (*i.e.,* radiographic SL angle and gap *Taleisnik, 1988*; *Boutin et al., 2014*) and *in silico* 3D SL gap computed across physiological wrist motions. Our findings suggest the novel scaffold for reconstruction of ruptured SLIL has potential to be a clinically

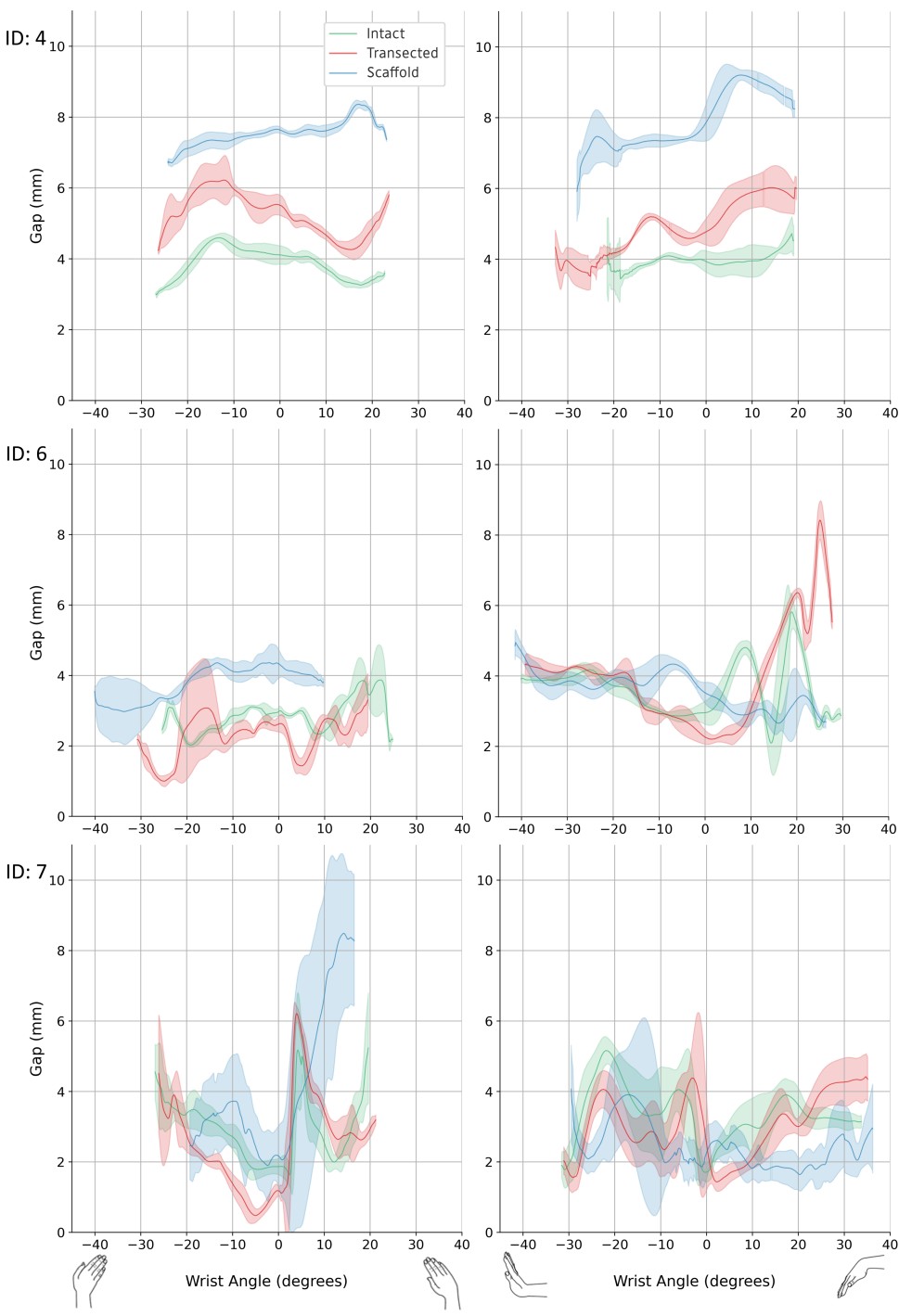

**Figure 7  3D SL gap.** Left: radial-ulnar deviation. Right: flexion-extension. Green: intact. Red: transected. Blue: scaffold. Graphs where data range varies along the *x*-axis was caused from any slippage of the robot-specimen fixation. 3D, three-dimension; SL, scapholunate.

**Table 2** Minimum and maximum 3D SL gap across each cycle for radial-ulnar deviation wrist motion, and at neutral pose.

| Sample number | Configuration | 3D SL Gap (mm) | | | | | | |
| --- | --- | --- | --- | --- | --- | --- | --- | --- |
| | | Minimum | | Maximum | | Delta | At neutral pose | |
| | | Mean | SD | Mean | SD | | Mean | SD |
| 4 | Intact | 2.93 | 0.05 | 4.63 | 0.12 | 1.70 | 4.11 | 0.25 |
| | Transected | 4.07 | 0.17 | 6.53 | 0.71 | 2.46 | 5.52 | 0.29 |
| | Scaffold | 6.62 | 0.08 | 8.43 | 0.11 | 1.81 | 7.64 | 0.09 |
| 6 | Intact | 1.87 | 0.12 | 4.71 | 0.19 | 2.84 | 2.98 | 0.12 |
| | Transected | 0.95 | 0.14 | 4.35 | 0.37 | 3.40 | 2.62 | 0.23 |
| | Scaffold | 2.56 | 0.59 | 4.70 | 0.37 | 2.14 | 4.33 | 0.40 |
| 7 | Intact | 1.39 | 0.27 | 6.56 | 0.48 | 5.17 | 1.87 | 0.23 |
| | Transected | 0.36 | 0.18 | 6.35 | 0.22 | 5.99 | 1.18 | 0.14 |
| | Scaffold | 0.82 | 0.21 | 10.76 | 1.04 | 9.94 | 2.17 | 0.85 |

Notes.
3D, three-dimension; SL, scapholunate; Delta, maximum minus minimum values.

**Table 3** Minimum and maximum 3D SL gap through each cycle for flexion-extension wrist motion, and at neutral pose.

| Sample number | Configuration | 3D SL Gap (mm) | | | | | | |
| --- | --- | --- | --- | --- | --- | --- | --- | --- |
| | | Minimum | | Maximum | | Delta | At neutral pose | |
| | | Mean | SD | Mean | SD | | Mean | SD |
| 4 | Intact | 3.23 | 0.26 | 7.01 | 1.55 | 3.78 | 3.93 | 0.30 |
| | Transected | 3.32 | 0.35 | 6.41 | 0.32 | 3.09 | 4.77 | 0.37 |
| | Scaffold | 3.60 | 1.25 | 9.43 | 0.11 | 5.83 | 7.86 | 0.53 |
| 6 | Intact | 1.80 | 0.61 | 6.28 | 0.83 | 4.48 | 2.95 | 0.33 |
| | Transected | 2.05 | 0.10 | 8.50 | 0.48 | 6.45 | 2.25 | 0.21 |
| | Scaffold | 2.19 | 0.41 | 5.66 | 0.96 | 3.47 | 3.53 | 0.28 |
| 7 | Intact | 1.31 | 0.39 | 5.40 | 0.20 | 4.09 | 1.71 | 0.28 |
| | Transected | 0.87 | 0.30 | 5.64 | 0.79 | 4.77 | 2.51 | 1.33 |
| | Scaffold | 1.00 | 0.16 | 6.45 | 1.98 | 5.45 | 2.26 | 0.74 |

Notes.
3D, three-dimension; SL, scapholunate; Delta, maximum minus minimum values.

viable solution as the SL linkage was not disturbed by its surgical installation, and in select cases provided limited kinematic restoration. However, the limited sample size in our study means results should not be generalized without confirmation in an expanded dataset. Issues of sample size aside, this preliminary analysis makes an important contribution to the literature as there is a complete lack of reports describing unloaded wrist kinematics in the presence of SL rupture.

Clinically, SLIL rupture is indicated by SL angle and gap quantified using radiographic imaging of the wrist. For an intact SLIL, typical values for the SL angle range 36 to 66°, while the SL gap is usually <2.0 mm (*Falck Larsen, Mathiesen & Lindequist, 1991*; *Chen et al., 2022*). We simulated these same SL clinical measures fusing motion capture with our 3D mesh models. In comparison to literature, the SL angle for the intact configuration was

lower in this study (this study range: 28.2–52.0°, mean: 41.9 ± 7.2°; Larsen et al. range: 36−66°, mean: 50.8 ± 6.7°). Post-transection, SL angle decreased an average of 2.3°, in contrast to literature reports of increased (>60°) SL angle following SLIL transection (*Chen et al., 2022*; *Gilula et al., 2002*). This discrepancy can be attributed to the minimally loaded state of the wrist in our study, whereas prior literature investigating dynamic instability typically loads the wrist considerably. After scaffold installation, SL angles for two of four specimens trended to and were below their intact values (*n*: 2, range: 27.3–32.9°, mean: 30.1–2.8°). In contrast to the SL angle, SL gap for the intact configuration (*n*: 9, range: 0.85–2.85 mm, mean: 1.73–0.58 mm) was within the range reported in the literature. Once transected, five of seven specimens demonstrated increase in SL gap (range: 2.03–5.09 mm, mean: 3.27–1.17 mm) indicative of SLIL deficiency. After scaffold installation, complete specimens had their SL gap restored to intact values (*n*: 3, intact mean: 1.41–0.16 mm, transected mean: 1.68–0.52 mm, scaffold mean: 1.52–0.26 mm). These results indicate that scaffold installation does not disturb the SL linkage, but the expectation of restoration of fully normal SL characteristics is not met.

In this study, dynamic stability of the SL was assessed computationally using a time-varying measure of 3D SL gap across robotically generated wrist motions. Compared to traditional two-dimensional SL gap, these measures are relatively large, explained by the third dimension, dynamic state, and approximated initial SL gap from scaffold placement. Consequently, change in gap range (*i.e.,* delta) was used to compare across configurations within a given specimen. For specimens 4 and 6, delta metric indicates the surgical introduction of the scaffold restored dynamic stability to the SL linkage (Tables 2 and 3). However, sample seven did not show this pattern of restoration, possibly due to poorly performing scaffold installation. These preliminary data suggest the scaffold is a promising surgical treatment for SLIL rupture, however, an expanded experimental dataset is needed to confirm these promising initial results.

One limitation of this study was the low magnitude radiocarpal compressive loading. Loading of this magnitude made comprehensive comparison with previous experiments challenging as prior studies often used larger compressive loading which has implications for soft tissue tension within the wrist and the resulting carpal kinematics. The 10.8 N of force proximally applied to the radiocarpal joint was intended to reduce the radiocarpal gap at extremes of manipulation, which it did effectively as evidenced by the joint remaining closed throughout robotic tests. A substantially greater axial load would likely affect carpal kinematics in all configurations; however, it is unclear whether this effect would be consistent across configurations and loading may have an interaction effect. Further, changes in carpal kinematics after transection of the dorsal radiocarpal ligament and SLIL have been reported previously (*Short et al., 2007*); however, we did not observe similar findings. This discrepancy may be attributed to differences in experimental setups and our limited sample size, limiting direct comparisons. Second, loads across the SL linkage were not measured within this study. It was infeasible to measure these loads due to the lack of physical space for instruments in addition to those used to track motion and support scaffold installation. Last, of the nine experiments originally conducted, three completed all

configurations. Therefore, statistical power is too low for any between-group comparisons to be made with confidence, hence we treat the results as hypothesis generating.

Several assumptions were also made during the study. First, the statistical shape model used to realign real bone posture to neutral assumed the novel bones (from cadavers) had shape that could be accurately modelled from the training set. We did not explicitly assess the shape model performance; however, shape reconstructions were statistically controlled to prevent ill-conditioned behaviour. Second, excessive tolerance between marker placement guide and specimen bones would result in play/slippage which would introduce errors into computational assessments. Future studies should use a more robust co-registration method for aligning physical markers to digital bones. One approach would be to rigidly fix a set of radiopaque markers to the specimens, so they can be recorded in CT scans and subsequently co-registered to motion capture data. Last, aligning FE and RUD axes of each specimen with the robot's x and y-axes when the wrist was in a neutral pose was performed manually. An automated method, supported by a robust co-registration method would be ideal (*e.g.*, incorporating a modern 3D electromagnetic tracking system).

## CONCLUSIONS

Results indicate surgical installation of the scaffold did not compromise the static and dynamic stability of the SLIL, with some indications of restoration towards native values. The scaffold shows promise in potentially providing sufficient mechanical fixation of the SL linkage and facilitating biological ligament ingrowth for treatment of SLIL injuries. Further research with larger sample sizes and enhancements in experimental setup are necessary to confirm or refute our results.

## ACKNOWLEDGEMENTS

The authors wish to thank those who donated their bodies to science so that this, and other, research projects could be performed. Results will further mankind's understanding of anatomy and can potentially improve patient care directly or indirectly.

### Funding

This work was funded by the Australian Medical Research Future Fund Scheme: BioMedTech Horizons 1.0. Administered body for funder: MTPConnect Award Number: SLIL_BMTH 07. Grant recipient: Prof David Lloyd. The funders had no role in study design, data collection and analysis, decision to publish, or preparation of the manuscript.

### Grant Disclosures

The following grant information was disclosed by the authors:
Australian Medical Research Future Fund Scheme: BioMedTech Horizons 1.0.
MTPConnect: SLIL_BMTH 07.

## Competing Interests

The authors declare there are no competing interests.

## Author Contributions

- Alastair R.J. Quinn conceived and designed the experiments, performed the experiments, analyzed the data, prepared figures and/or tables, authored or reviewed drafts of the article, and approved the final draft.
- Jayishni N. Maharaj performed the experiments, authored or reviewed drafts of the article, and approved the final draft.
- Randy Bindra conceived and designed the experiments, performed the experiments, authored or reviewed drafts of the article, and approved the final draft.
- Amelia Carr performed the experiments, authored or reviewed drafts of the article, provided literature/background information and sources, and approved the final draft.
- Natividad Gomez performed the experiments, authored or reviewed drafts of the article, provided literature/background information and sources, and approved the final draft.
- Kaecee Fitzgerald performed the experiments, authored or reviewed drafts of the article, provided literature/background information and sources, and approved the final draft.
- Nataliya Perevoshchikova performed the experiments, authored or reviewed drafts of the article, provided literature/background information and sources, and approved the final draft.
- Cedryck Vaquette conceived and designed the experiments, authored or reviewed drafts of the article, and approved the final draft.
- Claudio Pizzolato conceived and designed the experiments, authored or reviewed drafts of the article, and approved the final draft.
- Minghao Zheng conceived and designed the experiments, authored or reviewed drafts of the article, provided literature/background information and sources, and approved the final draft.
- David Lloyd conceived and designed the experiments, authored or reviewed drafts of the article, provided literature/background information and sources, and approved the final draft.
- David J. Saxby conceived and designed the experiments, analyzed the data, authored or reviewed drafts of the article, and approved the final draft.

## Human Ethics

The following information was supplied relating to ethical approvals (i.e., approving body and any reference numbers):

This study was approved by Griffith University's Human Research Ethics Committee (GU Ref No: 2018/533).

## Data Availability

The code is available at GitHub and Zenodo:

- https://github.com/a-quinn/SLIL_processing_public

- Alastair Quinn. (2025). a-quinn/SLIL_processing_public: Initial publication version (0.1). Zenodo. https://doi.org/10.5281/zenodo.15717632.

## Supplemental Information

Supplemental information for this article can be found online at http://dx.doi.org/10.7717/peerj.19766#supplemental-information.

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
