# Peer review of "Robotic evaluation of a 3D-printed scaffold for reconstruction of scapholunate interosseous ligament rupture: a biomechanical cadaveric study"

_PeerJ, doi:10.7717/peerj.19766_

## Round 0.1 · original submission · Major Revisions

Thank you for your submission. All four reviewers regard it to be well conceived and executed in most respects. One reviewer suggests Major Revisions and 2 suggest Minor Revisions. However, the 2 suggesting Minor Revisions have provided extensive reviews and/or comments to the manuscript file. While none of the concerns appear to be severely damaging to the manuscript, I opted to categorize it as Major Revisions for the number of suggestions. Please address these concerns and describe how you addressed them in your cover letter upon resubmission. I look forward to receiving your revised manuscript.

Reviewer 1 ·

Basic reporting

Quinn et. al. have investigated the effectiveness of a 3D printed PLA scaffold to to reconstruct a ruptured scapholunate interosseous ligament (SLIL) through robotic manipulation of human cadaveric wrists. The manuscript is written in clear and professional language. I appreciate the thorough discussion of the experimental setup and findings along with the appropriate context. The authors also acknowledge the limitations of their setup and provide helpful suggestions for those who seek to replicate or expand the study. The manuscript contains appropriate references and professionally made figures.

Experimental design

The scaffold investigated by Quinn et. al. has been validated in animal models but hasn't been studied in the context restoring of human SL kinematics. In order to study its applicability in humans, the authors set up an experiment with human cadaveric wrists with intact, transected and restored SLIL and manipulated them robotically to assess the restoration of the SL angle and gap after surgical implantation of the scaffold. The authors were able to identify restoration of the wrist kinematics, although with limited statistical power. Authors discuss the statistical limitations of the study in detail, which is appreciated. The authors also discuss the methods used, and limitations thereof in great detail allowing for appropriate replicability.

Validity of the findings

Authors conclude that the scaffold did not compromise the kinematic characteristics of the wrist and indicated some restoration potential. The authors successfully link the findings from their work to the original hypothesis and acknowledge all caveats such as need for larger sample sizes.

·

Basic reporting

- The article written clearly in easy to understand language.
- Are there any more recent references to support your statement in the introduction lines 60-62 that there is "no clinical consensus on best treatment"? The 2 references listed are over 30 years old.
- Can anything more be shared about the scaffold? It is frustrating that pictures of the scaffold are blurred out in figure 1 and not shown elsewhere. If some images and/or more descriptive text could be added it would be appreciated. If they cannot, please explicitly state the reason it cannot be shown.

Experimental design

- In line 123-124, why would the specimens not be thawed before imaging such that their position could be adjusted to a more standard position? Can you state an estimate of how much the shape model adjusted the relative position and orientation of the carpal bones, specifically the SL joint?
- during you initial assessment of potential marker occlusion (lines 167-168) what did you do if markers were occluded and do you think these adjustments resulted in markers loosening from bones and subsequently to specimens not making in through all test conditions?
- I am conflicted regarding the level of detail in section 2.3. From a clinical perspective I think it sufficiently explains what metrics were measured. That said, as an engineer who provides support to investigators trying to perform similar studies, not enough detail is provided such that I could recreate this method. A more explicit explanation of how the CT mesh models were aligned to the motion capture sensors and how anatomical the carpal kinematics were defined is needed to make such a recreation possible. If the target audience of this article/journal is the former, I understand if the authors' feel a more detailed explanation could make readers more versus less confused.

Validity of the findings

- I appreciate that the authors do not attempt to spin or hide the fact that the marker loosening was a problem in the study. That said, I also do not think the paper has sufficient evidence to make any statements on the biomechanical function of this scaffold. Given the lack of published literature on the SL joint kinematics following SLIL rupture, I think shifting the focus to how joint behavior changes following rupture would be good. Along these lines I am very interested in seeing more plots similar to those shown in figure 7 comparing the intact and transected conditions for the 8 available specimens. Additionally, the change ins Gap could be computed between these conditions with a mean delta gap shown.

Additional comments

As someone who has quantified carpal bone kinematics on cadaveric specimens, I appreciate the practical challenges associated with this study. I think the method and intact vs. transected findings are worthy of publishing without the addition of the scaffold condition. My overall recommendation would be that the authors either try to collect a few more specimens of data such that more conclusions can be drawn about the behavior of the scaffold and/or increase the focus on the findings of the other conditions.

Reviewer 3 ·

Basic reporting

Overall, I believe the manuscript is well written and easy to understand. I do have few questions which I have commented in the pdf. The overall idea of using a 3d printed scaffold was well explained with custom surgical guides. The figures are well made and supplementary data is sufficient for the manuscript.

Experimental design

Overall the methods are well defined but there's a scope of improvement which the authors allude to in the discussion. My major concern was the use of only 3 markers to track data when the tracking system (VZ10K) has 3 in-line cameras. With the acknowledgement that there's limited space and radius is fixed, having an additional marker for - scaphoid, lunate and metacarpal could have limited data loss.

Validity of the findings

The results are well written and are useful as preliminary data for hypothesis driven research question. Due to data loss for multiple reasons - there isn't enough power to compute meaningful statistical comparisons.

Additional comments

Overall, a very well conducted study for using a scaffold for reconstructing injured SLIL. With potential methodological improvements, there could be more meaningful data generated for confidence in the approach.

Annotated reviews are not available for download in order to protect the identity of reviewers who chose to remain anonymous.

Reviewer 4 ·

Basic reporting

The English is clear and professional in style. The list of references appears to be appropriate. Data is presented in graphical form, which is especially helpful in Figure 6. It is not presented in tabular form, which would be better suited to be primarily in the manuscript, and not as a supplement or excluded from the manuscript. There should be a clear statement of primary and secondary hypotheses in the Introduction that are subsequently tested in the Results and explained in the Discussion that is not present in the manuscript in current form.

Experimental design

See my comments on basic reporting? Why is the scaffold blurred in the photographs and insufficient detail in the Methods, Results, and Discussion? This is not acceptable for publication in this manner, transparency are a cornerstone of peer reviewed science.

Also, is there any consideration for the secondary stabilizers of the SLIL and their roles here for carpal kinematics. I would direct you to read and address the large body of work by Crisco and Wolfe.

Validity of the findings

It is difficult at this time for me to make an assessment that appropriate statistics were performed here.

Additional comments

None to add.

---

## Round 0.2 · accepted · Accept

Thank you very much for your detailed consideration of the reviewers' comments. The revised submission is suitably improved. Your manuscript is accepted. Please carefully proofread the page proofs when they are available. Congratulations!